# Monkeypox (Mpox) and Occupational Exposure

**DOI:** 10.3390/ijerph20065087

**Published:** 2023-03-14

**Authors:** Marta Szkiela, Marta Wiszniewska, Agnieszka Lipińska-Ojrzanowska

**Affiliations:** Nofer Institute of Occupational Medicine, Department of Occupational Diseases and Environmental Health, Sw. Teresy od Dzieciątka Jezus 8, 91-348 Lodz, Poland

**Keywords:** mpox, infectious diseases, occupational risk, occupational health, exposure, workplace, workers

## Abstract

Recently, there has been a significant increase in interest in biological risk factors, which are increasingly perceived as an important problem in occupational medicine. Exposure to harmful biological agents may be associated with the deliberate use of microorganisms in the work process or with unintentional exposure resulting from the presence of biological risk factors in the work environment. Monkeypox (mpox) is a viral infectious disease that may afflict humans and non-human primates. Since May 2022, mpox has occurred in Europe, North and South America, Asia, Australia and Africa, with some 76,713 cases (75,822 in locations that have not historically reported mpox) and 29 total deaths reported to date. Between 2018 and 2021, several cases of mpox were reported worldwide in high-income countries (Israel, Singapore, United Kingdom, United States: Texas and Maryland). We conducted a literature search in PubMed and Google Scholar web databases for occupational exposure to mpox. The highest work-related risk for mpox transmission has been noted among healthcare professionals, people working with animals, and sex workers. There is general agreement that a paramount issue to avoid transmission of infection in occupational settings is an appropriate decontamination of often-touched surfaces and usage of appropriate personal protective equipment by the workers at high risk of infection. The group that should especially protect themselves and be educated in the field of early symptoms of the disease and prevention are dentists, who are often the first to detect the symptoms of the disease on the oral mucosa.

## 1. Introduction

Recently, there has been a significant increase in interest in biological risk factors, which are increasingly perceived as an important problem in occupational medicine. Biological agents include viruses, bacteria, fungi and parasites. Despite significant technical progress and increased awareness of exposed people, biological risk factors still contribute to an increase in occupational risk in many occupational groups. Exposure to harmful biological agents may be associated with the deliberate use of microorganisms in the work process (e.g., in the biotechnology industry or in microbiological laboratories) or with unintentional exposure resulting from the presence of biological risk factors in the work environment (e.g., patients or biological material in healthcare facilities, waste or sewage in municipal management facilities and organic dust in agriculture) [1].

Monkeypox (mpox,) is a contagious disease, caused by the monkeypox virus (a member of the *Orthopoxvirus* genus in the family *Poxviridae*). This infectious disease can occur in humans and non-human primates. Mpox has small rodents as its natural reservoir and both humans and non-human primates are occasional hosts [2]. In the 1970s, the first case in humans was reported in the Democratic Republic of the Congo (DRC) in a 9-month-old boy, in a region where smallpox had been eliminated in 1968. Since 1970, human cases of mpox have been reported in 11 African countries: Benin, Cameroon, the Central African Republic, the DRC, Gabon, Cote d’Ivoire, Liberia, Nigeria, the Republic of the Congo, Sierra Leone and South Sudan [1]. Since May 2022, mpox has occurred also in Europe, North and South America, Asia, Australia and Africa, with some 76,713 cases (75,822 in locations that have not historically reported mpox) and a total of 29 deaths reports to date (as of October 2022) (Figure 1) [3].

On 28 November 2022, the WHO, after a series of consultations with world experts, decided to introduce a new preferred term “mpox” as a synonym for monkeypox (to avoid stigmatizing patients). Both names will be used simultaneously for a year while “monkey pox” will be phased out [4].

The most common signs and symptoms of mpox are: rash (98%), fever (67%), malaise (tiredness) (65%), chills (62%), pruritis (itching) (59%), headache (57%), enlarged lymph nodes (swollen glands) (57%), myalgia (56%), rectal pain (41%), rectal bleeding (23%), tenesmus (20%), pus or blood in stool (19%), vomiting or nausea (18%), proctitis (swelling, soreness in the rectal area) (16%), abdominal pain (15%), and conjunctivitis (redness or pain in the eye) (5%) [5]. Mpox symptoms usually start within 3 weeks of exposure to the virus. The illness typically lasts 2–4 weeks [5]. After the COVID-19 pandemic declaration, the World Health Organization (WHO) declared the mpox outbreak as a public health emergency of international concern [6]. One of the hypotheses explaining the current mpox outbreak is a general decline in the population immunity to smallpox and similar *Orthopoxvirus* diseases [7]. The smallpox vaccination program was discontinued 30 years ago, and during the current outbreak, a high incidence was recorded among adults who were unvaccinated against smallpox [7]. According to the WHO, vaccines used during the smallpox eradication program also provided protection against mpox [2]. 

Occupational exposure is most often defined as exposure to hazardous, harmful or strenuous work-related factors [8]. According to Polish law, the assessment of occupational exposure to biological risk factors takes into account the type of factor, determining the contact, latency period and determining the mechanism of action or the path of spread of the factor, without the need to determine the concentration of this factor [9]. According to Directive 2000/54/EC, biological agents include only cellular or non-cellular microbiological entities capable of replication and of provoking infection or other diseases. According to this directive, pathogens potentially present in the work environment have been divided into four risk groups—depending on the infectious properties they pose to an employee with a properly functioning immune system. The possibility of preventing and treating infections caused by them was also taken into account, and labels were introduced for those agents that can cause allergic effects (A) or synthesize toxins (T). The first group does not pose a risk of infection for humans, while pathogenic microorganisms have been classified into groups 2–4, with only viruses in the fourth group. Group 1 includes agents that are not likely to cause human disease. In the case of work with agents belonging to the first group of hazards, compliance with the rules of hygiene is a sufficient condition to eliminate exposure or limit the degree of exposure. Group 2 (e.g., *Borrelia burgdorferi, Clostridium tetani*) includes agents that can cause disease, group 3 (e.g., *Yersinia pestis, Mycobacterium tuberculosis*) includes agents that can cause severe disease and are a serious hazard to workers, but there is effective prophylaxis or treatment available for them. Group 4 (e.g., *Ebola Virus*) also includes agents that cause severe disease and are a serious hazard to workers, but there is usually no effective prophylaxis or treatment available for them [10].

According to a report by the European Agency for Safety and Health at Work, Biological agents and prevention of work-related diseases, five groups of high risk occupations were identified: animal-related occupations, waste and wastewater management, healthcare, arable farming and occupations that involve travelling for work and contact with travelers, such as for example in customs work [11].

The highest occupational risk associated with the mpox virus appears to apply to healthcare professionals (physicians, nurses, nursing assistants, emergency, medical technicians, therapists, pharmacists, students, laboratory workers), people working with animals (veterinarians, veterinary technicians, zoo keepers, employees of animal shops), and sex workers [12,13,14,15,16,17,18,19,20,21,22,23,24,25,26,27,28,29,30,31,32,33,34]. According to the US Centers for Disease Control and Prevention (CDC) data, mpox is more common in men than in women. These CDC data suggest that homosexual men, bisexual men, and other men who have sex with men make up the majority of cases in the current mpox outbreak [5].

## 2. Materials and Methods

We conducted a literature search between 1 October 2022 and 31 November 2022 in the MEDLINE web database (accessed using PubMed) and Google Scholar for the risk of mpox in occupational setting. The synthesis of searching terms was: “monkeypox” AND “occupational exposure”, “monkeypox in the workplace”, “occupational risk of monkeypox”. Studies aimed at the analysis of occupational exposure to mpox infections were included. The exclusion criterion was no occupational risk for mpox. Additionally, we screened the references of relevant systematic reviews and studies to identify further potentially eligible studies. No limits on the time of publication, language, or type of paper were set.

Our literature search resulted in 176 records, with 167 remaining after removing duplicates. Titles and abstracts were screened for inclusion (related to occupational exposure) and exclusion criteria (not related to occupational exposure). At this stage, 134 articles were rejected and 33 accepted for full text assessment. In total, 31 studies matched the selection criteria and were included in the final analysis. The literature search and selection process is demonstrated in Figure 2. 

## 3. Results

### 3.1. Healthcare Workers

Despite frequent occupational contact with patients or their biological samples, mpox infections among healthcare professionals (HCW) such as physicians, nurses, nursing assistants, paramedics, medical technicians, therapists, pharmacists, students, and laboratory workers are not common. In a study by Nörz D. et al. carried out in 2022, the areas in the rooms occupied by two patients with mpox on the fourth day of hospitalization were examined. The virus was found to be present on all surfaces directly touched by patients. The highest loads were found in the bathrooms. Mpox virus DNA was also found on surfaces of the patients’ rooms, on fabrics used by patients, and in hallways [13]. 

Fleischauer A.T. et al. conducted a study on the occupational mpox virus exposure of HCWs (those who entered a 2 m radius surrounding three patients with confirmed cases of mpox) during an epidemic in Wisconsin in 2003 [14]. Despite the fact that even 70% of HCWs had ≥1 unprotected exposures, none of them reported symptoms of mpox disease. One of the exposed HCWs, who had been vaccinated against smallpox in the last year, had serological evidence of a recent infection with the *orthopoxvirus*. The Advisory Committee on Immunization Practices has recommended preexposure vaccination for healthcare professionals since 1980, when smallpox was eradicated, due to the risk of *orthopoxvirus* infections transmission in occupational settings [15].

Vaughan et al. reported a case of mpox contracted by a healthcare professional in the workplace [16]. Transmission was likely through contact with contaminated bedding. Subsequently, 134 people potentially came into contact with an infected HCW, 4 of whom became ill. Initially, HCWs attending to the patient were equipped with standard personal protective equipment (PPE) consisting of disposable gowns and gloves. When a clinical diagnosis of mpox was suspected, high-consequence infectious disease (HCID) prevention and control measures were implemented (e.g., improved PPE consisting of a disposable gown, disposable gloves, three-filter mask and face shield or goggles). The patient was transferred to the Airborne HCID Treatment Center and mpox was confirmed by laboratory tests at Public Health England (PHE). Infection control precautions for contact persons (vaccination, daily monitoring, staying home from work) were implemented [16]. 

In the DRC, each year there are more than 1000 cases of human mpox, leading to exposures of healthcare workers. Petersen et al. analyzed data collected by the CDC from the ongoing CDC-supported enhanced surveillance program in the DRC Tshuapa Province [17]. Between 2010 and 2014, 1266 suspected cases of MPX were investigated in Tshuapa, 11 of which were HCWs. Taking into account all suspected cases between January 2010 and August 2014, the overall rate of HCW infections was 0.9% (range 0.6–1.8% per year). Among 699 confirmed cases in the analyzed period, 6 confirmed cases of mpox among HCWs accounted for 0.9% (range 0.3–3.1% per year). During the observation period, an average of 1.5 health professionals were infected per year, resulting in an estimated annual incidence rate of 17.4/10 000 [17]. 

Between 2018 and 2021, several cases of mpox were reported worldwide in high-income countries (Israel, Singapore, United Kingdom, United States: Texas and Maryland). All cases were travel related. Healthcare workers had contact with infected people, but none became infected [18,19,20,21,22]. During 1 May 2022–31 July 2022 in Colorado, a total of 313 Healthcare Personnel (HCP) were exposed to 55 patients with mpox. Seven HCP had exposure during aerosol generating procedures, three of whom wore an N95 respirator during their exposure. Overall, 273 (87%) exposures to patients with mpox rash or lesions occurred, and 161 (59%) included direct contact with the patient’s skin or lesions (gloves were worn in 125 exposures, were not worn in 30 exposures, and use of gloves was unknown for six exposures). Twenty-six (8%) exposed HCP reported handling linens; 23 (88%) of whom were wearing gloves. In Colorado, mpox transmission did not occur in the case of 313 HCP with varying levels of exposure to patients with mpox during patient care or through contaminated materials [23].

Among HCWs, laryngologists, dentists and dental assistants are one of the groups at risk of mpox, because primary changes in this disease, before appearing on the skin, are formed in the oropharynx. Perioral lumps with blisters and ulcers were initially reported during the current mpox epidemic [24]. Huhn G.D. found that nearly 30% of mpox patients reported mouth pain [25]. Dental care workers may be the first to detect the symptoms of mpox. They may be at risk from the production of droplets and aerosols during dental procedures and long-term close contact with patients. Fluid from skin or oral lesions containing mpox virus or from blood and saliva may be dispersed into the environment in the form of droplets and aerosols or as a result of direct contact with patients, creating a risk of exposure to occupational dental staff and hospital contamination of other patients [26]. Therefore, they should maintain a high degree of suspicion. Dental care workers must be alert to mpox-like symptoms and distinguish them from herpetic and similar vesiculo-blisters in the differential diagnosis. The primary mode of transmission of the virus is direct contact with skin lesions or with the patient’s personal belongings that have been in contact with the lesions. Therefore, in the dental care setting, transmission of infection can be prevented by using standard, contact, and drip-free precautions when treating patients with symptoms of mpox. Due to the potential risk of airborne transmission of the virus, airborne precautions should be taken according to the risk assessment and all dental staff should wear N95 masks, FFP3 respirators, fluid-resistant attire and eye protection [24].

### 3.2. Animal Workers (Abattoir and Slaughterhouse Workers, Agricultural Workers, Laboratory Workers, Veterinarians, Pet Shop Workers)

As mpox is also a zoonotic disease, veterinarians, zoo keepers, pet shop staff and other categories of workers who come into close contact with animals should be considered at high occupational risk of transmission.

In 2003, an outbreak of mpox occurred in Wisconsin among people who had contact with prairie dogs (staff of veterinary facilities where ill prairie dogs had received care, members of households with prairie dogs, pet store employees and visitors, animal distributors, visitor to a household with prairie dogs). Croft et al., in a cohort study, determined factors associated with occupational transmission. A standardized questionnaire was used as a research tool to determine prairie dog exposure, general occupational practices, demographic information, and medical history. Their investigation included active contact surveillance, exposure-related interviews, and a veterinary facility cohort study. They identified 19 confirmed, 5 probable, and 3 suspected cases. Rash, headache, sweats, and fever were reported by >80% of patients. Occupationally transmitted infections occurred in 12 veterinary staff, 2 pet store employees, and 2 animal distributors. Other cases occurred in six members of households with prairie dogs, four pet store visitors and one visitor to a household with prairie dogs. No known cases occurred in healthcare workers who treated patients or in laboratory workers who handled specimens [27]. 

According to Croft et al. [27], veterinary staff used PPE sporadically during high-risk activities. These findings underscore the importance of standard veterinary infection-control guidelines. The substantial amount of illness among veterinary staff underscores the importance of infection-control practices in veterinary settings. Cohort case-patients frequently did not use PPE during high-risk activities (e.g., examining or feeding ill prairie dogs). Furthermore, cohort members reported general work practices that foster hand-to-mouth activities in animal care areas. Only 12% of vets used gloves when cleaning ill animals’ cages, a task that can contaminate staff hands with animal dander, urine, and fecal matter. It is not possible to determine if infection control guidelines would prevent mpox infection among veterinary personnel, but use of PPE can reduce virus transmission [27].

In a study conducted by Dell et al., 292 women cooking for their households and 180 self-identified hunters from 21 villages bordering Murchison Falls National Park in Uganda were surveyed. The aim of the study was to understand bush meat preferences, the possibility of transmission of zoonotic pathogens, and awareness of common zoonoses related to wildlife. Both hunters and women who cook considered primates to be the most likely species of wild animals to transmit diseases that humans can catch. Among the common zoonotic pathogens, the highest percentage of cooking women and hunters believed that the pathogens causing abdominal pain or diarrhea and mpox could be transmitted by wild animals [28].

### 3.3. Sex Workers

A professional group that is difficult to study (due to its illegality in many regions of the world) are sex workers. Mpox is not classified as a sexually transmitted infection, but close skin-to-skin contact associated with oral, vaginal or anal sexual activity is a risk factor for the current mpox epidemic. As much as 99% of reported US mpox cases afflicted men, among whom 94% admitted male-to-male close intimate or sexual contact [29]. At the end of April 2022, mpox virus transmission during condomless sexual intercourse has been documented in the UK in two white British men. Patients did not report any recent travel to regions with endemic mpox or outside the UK. The location of the primary lesions corresponded to sexual contact [30]. In Germany, the first two human cases of mpox infection were diagnosed in two men who had sex with men. One of the patients declared that he provided sexual services to men [31].

In Italy, four cases of mpox have been reported among young adult men reporting sexual intercourse (with men) without a condom. They all traveled in the first 2 weeks of May 2022. Three of them attended a mass event on the island of Gran Canary and one was traveling for the purpose of providing sexual services. All admitted having had condomless sex with various male partners during the trip [32]. A study by Thornhill et al. found that mpox DNA was detected in the seminal fluid of 91% of those affected by monkeypox in a large case series during the 2022 global outbreak. It was suspected that 95% of people with the infection contracted it through sexual activity. In this case series, 95% of patients developed rash, 73% had anogenital lesions and 41% had mucosal lesions. Concomitant sexually transmitted infections were reported in 29% of those tested [33]. In September 2022, a case report was published of mpox in a man who self-identified as a sex worker. The patient was also a carrier of the HIV virus. He shed mpox virus DNA in his semen for more than 3 weeks [34].

Sex work often involves long-lasting and extensive face-to-face, skin-to-skin, mucosa-to-mucosa (e.g., oral, genital, or anal) contact with their clients. Objects that have come into contact with a lesion such as towels, bedding or clothing can serve as fomites [36].

### 3.4. Prevention Methods

An Italian study conducted among 163 general practitioners, public health specialists and occupational physicians in May 2022 showed that the state of knowledge about mpox infection is unsatisfactory, with significant gaps in knowledge about all aspects of the disease and the significant omission of mpox as a pathogen, especially when compared to SARS-CoV-2, TB, HIV and HBV [37]. Medical professionals may face some difficulties in diagnosing and treating mpox, which is a fundamental part of a prevention strategy. A Czech study conducted among healthcare workers in September 2022 showed that only 8.8% of the respondents agreed to be vaccinated against HMPXV, almost half of the respondents (44.9%) stated that they would not be vaccinated, and 46.3% were hesitant. It raises concern that among the most frequently used sources of information about HMPXV were digital news portals (47.5%) and social media (25.8%), while scientific journals (5.6%), the ECDC (5%) and CDC in the USA (1.5%) were consulted the least [38].

Workers at risk should be advised on self-management, isolation and prompt reporting of symptoms. Activities where the worker may be exposed to the aerosol, such as shaking sheets, have to be considered as high risky. Exposed workers to mpox should undergo a 21-day active surveillance including the incubation period. Post-exposure vaccinations may be administered with available varicella vaccines, preferably within 4 days of exposure and up 14 after exposure [17,18,19,20,21]. 

Following the recent outbreak of mpox, disease control centers in both the US and Europe have provided guidance on protective measures to be followed in healthcare facilities to reduce the risk of contracting and spreading the mpox infection. There is general agreement that appropriate personal protective equipment (PPE) is needed, including a filtering respirator with a filter tested for fit (for example, an N95 respirator) or a powered air-purifying respirator, gloves and aprons [39].

Koenig et al. presented a framework tool that can be used by health policy makers [40,41]. It is based on the “identify–isolate–inform” procedure: patients are identified as potentially at risk or infected after an initial assessment of risk factors and accompanying symptoms. Employees must put on PPE and isolate infected patients immediately, and information exchange with the agencies’ infection control offices must also take place as soon as possible [40,41].

## 4. Conclusions

The recent COVID-19 pandemic and the mpox outbreak have shown that urgent measures are needed to protect workers from the effects of animal-to-human and human-to-human transmission of infectious diseases. These outbreaks have also shown that such exposure can affect a wide range of occupations, although this may not have been recognized at first.

Occupational exposure to mpox has recently become a cause for concern. Healthcare staff, especially dentists and laryngologists, professionals who work with animals and sex workers seem to be under the highest work-related risk for mpox transmission. Education about early symptoms recognition, the necessity of patient isolation and proper usage of appropriate personal protective equipment by the professionals, play a key role in mpox prevention. 

Both the development of research techniques in the field of microbiology, molecular biology and immunology, as well as the emergence of new species or varieties of microorganisms that cause dangerous infectious diseases in various parts of the world mean that new biological agents posing an occupational hazard are detected every year. The views on epidemiology and the importance of specific factors in particular work environments are also being modified. This creates the need to undertake appropriate research and activities regarding the recognition, monitoring and prevention of specific biological risk factors and diseases caused by them.

Due to the importance of the complex issue of the biological factors of occupational hazards, the following courses of action seem appropriate [42]:Equipping employees with personal protective equipment that effectively protects their respiratory system (modern respirators with forced air flow), mucous membranes and skin (new generation suits);Providing exposed employees with professional medical care, ensuring the performance of appropriate examinations and preventive vaccinations;The use of special security systems in healthcare facilities and laboratories where exposure to highly infectious microorganisms may occur;Developing health education, which in the opinion of specialists is the cheapest and at the same time the most effective means of protection against biological agents.The creation of monitoring systems that collect notifications of such diseases. According to the European Agency for Safety and Health at Work report, Biological agents and prevention of work-related diseases [11], a number of monitoring systems that collect notifications of such diseases exist, mostly in the area of public health, but the information was not centralized and therefore not easily accessible. There is also a missing link to occupational safety and health.

Our publication has some strengths and limitations. The strength of this review is that it included a broad search strategy on the occupational risk of monkeypox worldwide, without time or language limits, which reduced selection bias. Overall, 28 relevant sources were identified for comprehensive data extraction. There are also limitations. The monkeypox epidemic is a relatively new phenomenon, there are not many studies on this topic, especially studies on occupational exposure.

## Figures and Tables

**Figure 1 ijerph-20-05087-f001:**
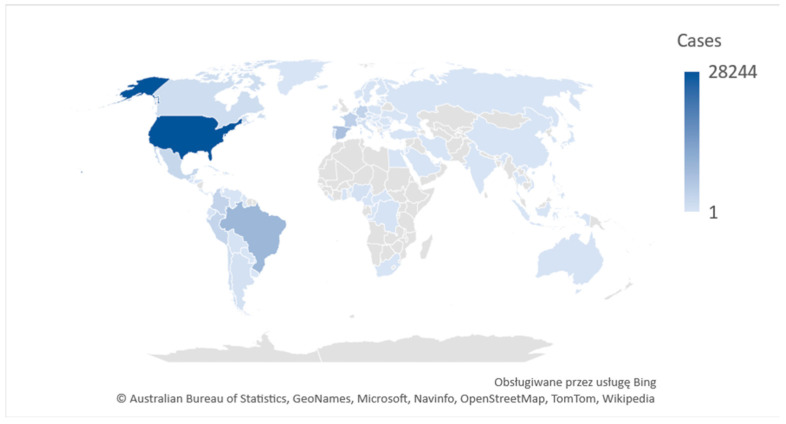
2022 Mpox Outbreak Global Map. Own graphics based on WHO [4].

**Figure 2 ijerph-20-05087-f002:**
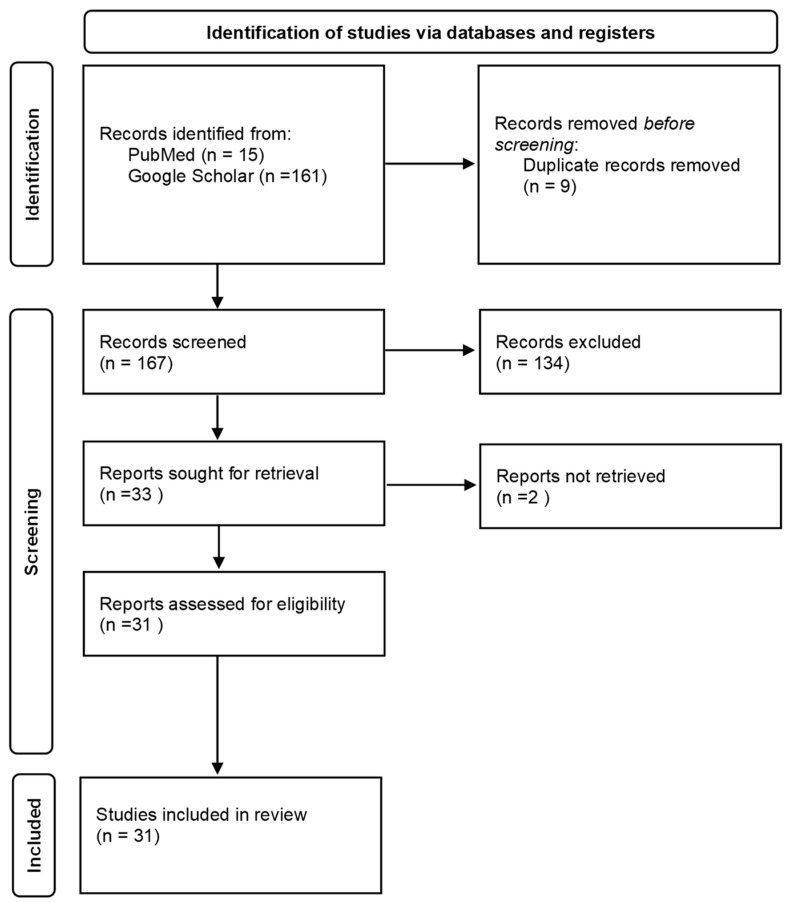
Flow chart of the selection process [35].

## Data Availability

Not applicable.

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
