# Peer review of "Monkeypox (Mpox) and Occupational Exposure"

_ijerph, 2023, doi:10.3390/ijerph20065087_

Round 1

Reviewer 1 Report

In this manuscript, Szkiela et al. reviewed occupational factors and risks linked to infection and transmission of monkeypox virus with emphasis on the current outbreak. The study has merit. However, more clarifications are necessary. Here I have raised a few points for the authors to revise the manuscript and hopefully improve the quality of the manuscript:

Minor points

1.    It is commendable that the review has the new nomenclature suggested by the WHO for the disease: mpox. However, I do recommend that authors standardize the manuscript with a single term for the disease. Although they are synonymous, the new term "mpox" is preferred. In addition, the authors present the acronym "MPX" but continue to use the 3 terms throughout the manuscript. If you choose "mpox", I believe that the abbreviation "MPX" is not necessary.

2.    I suggest removing the quotation marks from the words sex workers throughout the manuscript.

3.    Page 1 – line 25 – “mpox virus”. The naming of virus is the responsibility of the International Committee on the Taxonomy of Viruses (ICTV), so the most correct way now is to keep the name as monkeypox virus. Authors can consult the ICTV: https://ictv.global/

4.    Page 1 – lines 26-27 “This infectious disease can occur in humans and non-human primates”. Add a reference to this sentence. In addition, it is important to emphasize that this is a virus with a broad host spectrum, where the role of rodents as natural reservoirs is well described, it is not a virus that only infects primates.

5.    Was figure 1 based on reference 2? Please make this clear in the text. In addition, authors must cite Figure 1 at some point in the text.

6.    Please check if the resolution of the figures is in accordance with the journal's standards.

7.    Page 4 – Lines 96 and 98. “Orthopox virus”. Please change the word to “orthopoxvirus” as it refers to the genus

Author Response

Dear Reviewer,

Thank you for all your work on our manuscript “Monkeypox (Mpox) and occupational exposure.” Your comments and suggestions were very useful and helped to improve the paper considerably. All your suggestions have been taken into account in the recent revision of the manuscript. Below you can find answers to your specific comments.

Reviewer 2 Report

1. In the entire manuscript, names of species and genera must be italicized.
Please follow the instructions below: https://wwwnc.cdc.gov/eid/page/scientific-nomenclature

2. As you said in lines 36-39, the current official name of the monkeypox is MPOX. Please harmonize terminology across the entire manuscript.

3. Line 56-59: this important statement must have a relevant reference.

4. Line 59: remove parentheses around sex workers.

5. Line 65: When did you perform the search?

6. Add a definition for "occupational exposure/risk" either in the Introduction section or the Materials & Methods.

7. What was your synthesis method? Meta-analysis? Thematic analysis? Meta ethnography? or what exactly?

8. Line 82 -85: this paragraph is unclear.

9. Line 134 - 152: you narrative may benefit from reflecting further on the oral manifestations of MPOX that can help dentists to figure them out earlier.

10. Line 199-222: as long you have mentioned the prevention methods, it is vital to reflect on the role of HCWs in the context of primary prevention immunization.
Suggested ref: https://pubmed.ncbi.nlm.nih.gov/36560432/

11. Please add the limitations, strengths, and implications of your review.

12. Knowledge of HCWs and their vaccination perceptions towards MPOX are also vital for your discussion narrative.

Sincerely,

Author Response

(The authors gave the same response as above.)

Reviewer 3 Report

Dear author

I apreciate your efforts to publish your draft titled "Monkeypox (Mpox) and occupational exposure". My main concern is related to the methodology for the documents searching has been clarified and the research design improved.

The actual research design does not consider reproducible systematic review approach.  I consider important biases related to this and my recomendation is reject the paper related to this important flaws.

I recommend changes and major clarifications in the remarks that I add in the draft in the attached .pdf file.

Author Response

Dear Reviewer,

Thank you for all your work on our manuscript “Monkeypox (Mpox) and occupational exposure.” Your comments and suggestions were very useful and helped to improve the paper considerably. All your suggestions have been taken into account in the recent revision of the manuscript. 

Reviewer 4 Report

It is great to summarize individually what other studies have reported so far but it is not clear what is the goal of this manuscript. It is hard to catch the overall aim of this manuscript and what the authors project or conclude from each reference they are summarizing. It would be better if there is actual discussions and insight across all the reported cases together instead of simply summarizing each publication.

In the conclusion section, the authors mentioned the 'highest risk' without any evidence based on statistics. It would be more accurate to say A, B, and C has been reported for such infection, etc.

Author Response

(The authors gave the same response as above.)

Round 2

Reviewer 2 Report

References list is incomplete in the revised version.

Author Response

(The authors gave the same response as above.)

Reviewer 3 Report

Dear authors

We appreciate your efforts to improve your manuscript.

However, almost none of my previous redactions were taken into account and they are repeated in the new one in the manuscript that they have sent us.

I recommend taking my suggestions into account and rethinking your methodology to improve your search strategy and resubmit your article again.

Author Response

(The authors gave the same response as above.)

Reviewer 4 Report

The manuscript has been improved and I have no further suggestions.

Author Response

(The authors gave the same response as above.)
